# Click Chemistry of Melamine Dendrimers: Comparison of “Click-and-Grow” and “Grow-Then-Click” Strategies Using a Divergent Route to Diversity

**DOI:** 10.3390/molecules28010131

**Published:** 2022-12-23

**Authors:** Sanami Numai, Risako Yoto, Masataka Kimura, Eric E. Simanek, Yoshikazu Kitano

**Affiliations:** 1Laboratory of Bio-Organic Chemistry, Tokyo University of Agriculture and Technology, Tokyo 183-8509, Japan; 2Department of Chemistry & Biochemistry, Texas Christian University, Fort Worth, TX 76129, USA

**Keywords:** dendrimer, triazine, click chemistry, Huisgen cycloaddition

## Abstract

Dendrimers are attractive macromolecules for a broad range of applications owing to their well-defined shapes and dimensions, highly branched and globular architectures, and opportunities for exploiting multivalency. Triazine dendrimers in particular offer advantages such as ease of synthesis, stability, well-defined spherical structure, multivalency, potential to achieve acceptable drug loadings, and low polydispersity. In this study, the potential utility of alkyne-azide “click” cycloadditions of first-, second-, and third-generation triazine dendrimers containing three or six alkynyl groups with benzyl azide was examined using copper catalysts. “Click-and-grow” and “grow-then-click” strategies were employed. For the first- and second- generation dendrimers, the desired triazole derivatives were obtained in high yields and purified by simple reprecipitation without column chromatography; however, some difficulties were observed in the preparation of third-generation dendrimers. The desired reaction proceeded under microwave irradiation as well as with simple heating. This click chemistry can be utilized for various melamine dendrimers that are fabricated with other amine linkers.

## 1. Introduction

Dendrimers are attractive macromolecules for a broad range of applications owing to their well-defined shapes and dimensions, highly branched and globular architectures, and opportunities for exploiting multivalency [1,2,3]. Dendrimers can also be modified by an introduction of various functional groups, which is a promising strategy for enabling their use in drug delivery vehicles and catalysis [4,5,6,7,8,9,10,11,12,13]. For example, triazine dendrimers are a class of dendrimers that offer notable advantages such as ease of synthesis, stability, well-defined spherical structure, multivalency, potential to achieve acceptable drug loadings, and low polydispersity [14,15]. Recently, Simanek et al., synthesized various triazine dendrimers with multiple functional groups such as hydroxy and amino groups at the end of the molecule, and modified each functional group to incorporate multiple functionalities into the triazine dendrimers [16,17,18]. To further expand the potential of triazine dendrimers, Huisgen cycloaddition reaction was employed. Huisgen cycloaddition is a typical click reaction that affords 1,2,3-triazoles by the cycloaddition of azides and alkynes. Using this reaction, various substituents can be readily introduced into molecules with high functional selectivity without significant by-products. In recent years, click chemistry has been applied to the syntheses of dendrimers. Expanding the use of click reactions to dendrimers can contribute to the development of dendrimer science [19,20,21,22,23,24]. However, the application of Huisgen cycloaddition to triazine dendrimers has not been reported to date. Herein, synthesis of triazine dendrimers with alkyne chains and a molecular modification of triazine dendrimers using Huisgen cycloaddition is reported. For this reaction, the traditional piperazine linker was selected to obtain the triazine dendrimer. Subsequently, two strategies, namely “click-and-grow” and “grow-then-click,” were proposed for the synthesis and molecular modification of the triazine dendrimers. The “click-and-grow” strategy allows the generation of radial diversity by employing different azides at each generation, while the “grow-then-click” strategy relies on carrying alkynes through the synthesis and performing click reactions globally. After each generation of dendrimers was synthesized, all alkyne side chains were subjected to Huisgen cycloaddition.

## 2. Results and Discussion

The “click-and-grow” strategy to prepare the first-generation (G1) dendrimer **3** containing three alkynes is outlined in Figure 1. A stoichiometric amount of monochlorotriazine **2** [17] was treated with tris(piperazyl) triazine core **1** [25] in the presence of excess base for three days by refluxing CHCl_3_ to afford G1 dendrimer **3** in 91% yield. The desired dendrimer **3** was easily purified from unreacted **2** and incompletely substituted cores by silica gel column chromatography.

A well-known catalyst, CuSO_4_/ascorbic acid [26], was initially used for the Huisgen cycloaddition reaction of G1 dendrimer **3** with benzyl azide at room temperature. The progress of the reaction was monitored using thin layer chromatography (TLC) and mass spectrometry; the reaction remained incomplete after three days. This low reactivity was likely due to the solubility of dendrimer **3**. While this catalytic reaction is typically performed in an aqueous solution, dendrimer **3** did not show sufficient solubility in mixed solvent systems (THF/water). The same cycloaddition reaction of G1 dendrimer **3** was performed with CuI as the copper catalyst in THF. However, the reaction remained incomplete after three days, but the solubility of dendrimer **3** was improved. In addition, a byproduct with 5-iodo-1,2,3-triazole ring [27,28] was generated. This cycloaddition of G1 dendrimer **3** was optimized using microwave irradiation based on the previously published reports; [29] the results are summarized in Table 1. The desired triazole dendrimer **4** was obtained in 85% yield when the reaction was performed with CuSO_4_/ascorbic acid in THF/water under microwave irradiation for 15 min (Entry 1). The remaining copper salts were easily removed by washing with aqueous NaOH solution. The desired product was obtained by simple reprecipitation with MeOH from a clear solution of the crude product in CHCl_3_. Dendrimer **4** was obtained in 94% yield when the reaction was performed with CuI in THF under identical microwave irradiation conditions (Entry 2). The byproduct with the 5-iodo-1,2,3-triazole ring was not generated in this case. These results suggested that microwave irradiation significantly improved the yield of the cycloaddition reaction of the triazine dendrimer. The same reaction was carried out using a pressure vessel in an oil bath at 110 °C without microwave irradiation. The desired reaction occurred, affording comparably high yields (Entries 3 and 4), albeit more slowly.

The synthesis and modification of the second-generation dendrimer **6** is shown in Figure 2. The deprotection of the Boc groups of dendrimer **4** was achieved with 50% trifluoroacetic acid (TFA) in CH_2_Cl_2_. Dendrimer **5** was extracted using CHCl_3_ from a basic solution of NaOH, and then used without further purification. This material was treated with a stoichiometric amount of monochlorotriazine **2** in the presence of excess base for five days under reflux conditions to afford **6** in 97% yield. Dendrimer **6** was easily purified by silica gel column chromatography. Copper-catalyzed alkyne-azide cycloaddition reaction of **6** with benzyl azide was examined under microwave irradiation with CuSO_4_/ascorbic acid and CuI, and the desired triazole dendrimer **7** was obtained in 91% yields, respectively (Entries 5 and 6). When the reactions were performed at 110 °C in a pressure vessel, the desired triazole dendrimer **7** was also obtained in high yield (Entries 7 and 8). However, a longer reaction time (2×) was required to complete the reaction compared to the G1 dendrimer **3**. G2 dendrimer **7** was purified through simple precipitation by MeOH addition to the crude organic phase obtained from extraction as well as G1 dendrimer **4**.

The successful cycloaddition-mediated derivatization of **6** led us to investigate the click chemistry of a third-generation dendrimer. Dendrimer **9** was prepared via an iterative extension of **7** (Figure 3) and purified using silica gel column chromatography. The copper-catalyzed alkyne-azide cycloaddition reaction of **9** with benzyl azide was investigated under microwave irradiation using CuSO_4_/ascorbic acid and CuI; the desired triazole dendrimer **10** was obtained with low yields of 11% and 14%, respectively (Entries 9 and 10). When the reactions were performed at 110 °C in a pressure vessel, the desired triazole dendrimer **10** was also obtained (Entries 11 and 12). The click modification of G3 dendrimer **9** required a longer reaction time than that in the case of G2 dendrimer **6** for the complete disappearance of **9**. TLC analysis showed evidence for the formation of polar, potentially polymeric species that could arise from alkyne–alkyne homocoupling reactions [30]. In addition, triazole dendrimer **10** could not be purified by simple precipitation because of its solubility limitations in various solvents and the presence of multiple impurities. Dendrimer **10** was purified using silica gel column chromatography.

Although not explored in the syntheses described above, the “click-and-grow” strategy allows the incorporation of different azide-bearing groups at each generation of the dendrimer. Alternatively, a single azide-bearing group can be incorporated throughout the dendrimer if pendant alkynes are carried through the iterative growth of these targets and globally “clicked.” To explore the “grow-then-click” strategy, second- and third-generation dendrimers having six and nine alkynyl groups in the molecules, respectively, were targeted for synthesis. Dendrimer **12** was prepared from **3** in 76% overall yield using a similar method as that employed for preparing G2 dendrimer **6** (Figure 4). The reaction of **12** with benzyl azide under microwave irradiation with CuSO_4_/ascorbic acid and CuI yielded the desired triazole dendrimer **7** in high yields (Entries 13 and 14). When the same reactions were carried out at 110 °C in a pressure vessel, the desired triazole dendrimer **7** was also obtained in high yields (Entries 15 and 16). The yields and reaction times of the cycloaddition reaction of dendrimer **12** were similar to those of dendrimer **6**. However, the preparation of dendrimer **14** was more challenging than the synthesis of G2 dendrimer **12** (Figure 5). Although deprotection of **12** was achieved with 50% TFA in CH_2_Cl_2_, the low solubility of **13** in organic solvents prevented the synthesis of **14**. TLC analysis indicated that a significant amount of **2** remained after one week. Notably, dendrimer **14** could not be synthesized even after changing the solvents to CHCl_3_/MeOH (5:1), CHCl_3_/THF, THF, dichloroethane, and dioxane. The poor solubility of dendrimer **13** led to the abandonment of this route.

An alternative approach to prepare **14** was attempted. The reaction of G1 dendrimer **11** with second-generation dendron **17** was envisioned to afford G3 dendrimer **14**. Dendron **17** was prepared from **2** in 65% overall yield (Figure 6). Monochlorotriazine **2** was treated with excess piperazine, which afforded mono-*N*-substituted piperazine **15**. Subsequently, **15** was converted into dichlorotriazine **16** with excess cyanuric chloride. The by-products of the reaction with excess reactants were observed in both cases. A stoichiometric amount of dendron **17** was treated with G1 dendrimer **11** for one week under reflux conditions using CHCl_3_/THF (1:1) mixture. In this case, the reaction mixture was a clear solution at the beginning of the reaction, but precipitation gradually increased in the mixture. However, TLC analysis showed the presence of a large amount of remaining **17**, while the desired G3 dendrimer **14** was not observed. These results suggested that the synthesis of dendrimer **14** was difficult because of solubility issues.

Additionally, the synthesis of G3 triazine dendrimers **9** and **10** with 1-benzyl-1,2,3-triazole rings was accomplished. This result led us to attempt a new strategy for synthesizing G3 dendrimer **18** with three 1-benzyl-1,2,3-triazole rings. Dendrimer **18** was prepared via an iterative extension of **17** (Figure 7) with dendrimer **5** and purified using silica gel column chromatography. The copper-catalyzed alkyne-azide cycloaddition reaction of **18** and benzyl azide was investigated under microwave irradiation with CuSO_4_/ascorbic acid and CuI. The desired triazole dendrimer **10** was obtained with low yields of 11% (with CuSO_4_/ascorbic acid) and 10% (with CuI) (Entries 17 and 18). When the reactions were performed at 110 °C in a pressure vessel, the desired triazole dendrimer **10** was also obtained (Entries 19 and 20).

## 3. Materials and Methods

### 3.1. General Experimental

All chemicals were obtained from TCI Fine Chemicals, Tokyo, Japan, Wako Pure Chemical Industries, Tokyo, Japan, Kanto Chemical, Tokyo, Japan and Sigma-Aldrich, St. Louis, MO, USA, and used without further purification. NMR spectra were recorded in CDCl_3_ or CDCl_3_/CD_3_OD (5:1) on either a JEOL ECS-400 or a JEOL ECA-600 spectrometer, Tokyo, Japan. ^1^H NMR data are reported in ppm (δ) relative to TMS. ^13^C NMR data are reported in ppm (δ) relative to the central line of the triplet for CDCl_3_ at 77.0 ppm. Mass spectra were recorded on a JEOL JMS-S3000 SpialTOF instrument, Tokyo, Japan. Microwave experiments were carried out using a CEM Discover Microwave Synthesizer (CEM Corporation, Tokyo, Japan), and the irradiation was performed at a maximum power of 150 W. Chromatographic separations were carried out on a silica gel column (Kanto Chemical 60N, 63–210 µm, Tokyo, Japan; or Chromatorex^®^ NH-DM1020, 100–200 mesh, Fuji Silysia Chemical Ltd., Tokyo, Japan). The NMR spectra are shown in Appendix A.

### 3.2. Synthesis of G1 Dendrimer ***3***

Compound **2** (4.75 g, 13.5 mmol) and DIPEA (7.1 mL, 41.8 mmol) were successively added to a solution of triazine core **1** (1.37 g, 4.11 mmol) in CHCl_3_ (30 mL), and the resulting mixture was refluxed for three days. After the reaction mixture was concentrated under reduced pressure, the residue was dissolved in CHCl_3_ (70 mL). The organic phase was washed with water (50 mL × 3), dried over Na_2_SO_4_, and concentrated under reduced pressure. The crude product was purified by silica gel column chromatography (gradient elution using CH_2_Cl_2_/EtOAc (2:1) until no detectable **2** was observed, as determined by UV spotting, to CHCl_3_/MeOH (10:1) to obtain the desired product) to afford **3** as a white solid (4.82 g, 91%). ^1^H NMR (600 MHz, CDCl_3_) δ 4.94 (t, *J* = 5.7 Hz, 3H), 4.19 (dd, *J* = 5.7, 2.5 Hz, 6H), 3.85–3.71 (m, 36H), 3.48–3.42 (m, 12H), 2.20 (t, *J* = 2.5 Hz, 3H), 1.49 (s, 27H). ^13^C NMR (150 MHz, CDCl_3_) δ 165.8, 165.4, 165.2, 154.8, 80.8, 79.9, 70.7, 43.1, 43.0, 42.9, 30.4, 28.4. HRMS (MALDI): calcd for C_60_H_87_N_27_NaO_6_, 1304.7225 (M + Na^+^); found, 1304.7240.

### 3.3. Synthesis of G1 Dendrimer ***4***

#### 3.3.1. CuSO_4_/Ascorbic Acid

Benzyl azide (33.0 µL, 0.264 mmol), ascorbic acid (21 mg, 0.119 mmol), and copper (II) sulfate (3.7 mg, 0.0232 mmol) were successively added to a solution of compound **3** (101 mg, 0.0787 mmol) in THF/H_2_O (1:1 *v*/*v*, 5 mL). The resulting mixture was subjected to microwave irradiation at 110 °C for 15 min in a sealed vial or stirred at 110 °C for 1 h in a pressure vessel. After the reaction mixture was concentrated, the residue was dissolved in CHCl_3_ (70 mL), and the solution was washed with 5% aqueous NaOH (30 mL × 2) and water (30 mL × 2). The organic phase was dried over Na_2_SO_4_ and then evaporated. The crude product was purified by reprecipitation with MeOH from a clear solution of CHCl_3_ to afford **4** as an off-white solid. Yield; MW: (113 mg, 0.0672 mmol, 85%), pressure vessel: (110 mg, 0.0654 mmol, 83%).

#### 3.3.2. CuI/DIPEA

Benzyl azide (33.0 µL, 0.264 mmol), DIPEA (40 µL, 0.235 mmol), and copper (I) iodide (4.9 mg, 0.0257 mmol) were successively added to a solution of compound **3** (101 mg, 0.0787 mmol) in THF (5 mL). The resulting mixture was subjected to microwave irradiation at 110 °C for 15 min in a sealed vial or stirred at 110 °C for 1 h in a pressure vessel. After the reaction mixture was concentrated, the residue was dissolved in CHCl3 (70 mL) and the solution was washed with 5% aqueous NaOH (30 mL × 2) and water (30 mL × 2). The organic phase was dried over Na_2_SO_4_ and then evaporated. The crude product was purified by reprecipitation with MeOH from a clear solution of CHCl_3_ to afford **4** as an off-white solid. Yield, MW: (125 mg, 0.0743 mmol, 94%); pressure vessel: (118 mg, 0.0702 mmol, 89%). ^1^H NMR (600 MHz, CDCl_3_) δ 7.44–7.29 (m, 12H), 7.27–7.20 (m, 6H), 5.48 (s, 6H), 5.23 (t, *J* = 6.1 Hz, 3H), 4.67 (d, *J* = 6.0 Hz, 6H), 3.83–3.65 (m, 36H), 3.49–3.32 (m, 12H), 1.49 (s, 27H). ^13^C NMR (150 MHz, CDCl_3_) δ 166.1, 165.4, 165.2, 154.8, 146.9, 134.6, 129.1, 128.8, 128.0, 121.5, 79.9, 54.1, 43.0, 43.0, 42.9, 36.6, 28.4. HRMS (MALDI): calcd for C_81_H_108_N_36_NaO_6_, 1703.9145 (M + Na^+^); found, 1703.9137.

### 3.4. Synthesis of G1 Dendrimer ***5***

TFA (10 mL) was added to a solution of compound **4** (309 mg, 0.184 mmol) in CH_2_Cl_2_ (10 mL), and the resulting mixture was stirred for 3 h at room temperature. After the reaction mixture was concentrated under reduced pressure, the residue was basified with 20% aqueous NaOH (50 mL). The aqueous solution was extracted with CHCl_3_ (50 mL × 6). The organic phase was washed with water (30 mL × 1), dried over Na_2_SO_4_, and concentrated under reduced pressure to afford **5** as a white solid (228 mg, 90%), which was used for the next reaction without further purification. ^1^H NMR (600 MHz, CDCl_3_) δ 7.39–7.31 (m, 12H), 7.26–7.20 (m, 6H), 5.47 (s, 6H), 5.24 (t, *J* = 6.1 Hz, 3H), 4.67 (d, *J* = 6.1 Hz, 6H), 3.96–3.55 (m, 39H), 2.88–2.79 (m, 12H).^13^C NMR (150 MHz, CDCl_3_) δ 166.1, 165.3, 165.2, 165.1, 147.0, 134.6, 129.1, 128.7, 128.0, 121.5, 54.1, 46.0, 44.2, 43.0, 43.0, 36.6. HRMS (MALDI); calcd for C_66_H_84_N_36_Na, 1403.7572 (M + Na^+^); found, 1403.7561.

### 3.5. Synthesis of G2 Dendrimer ***6***

Compound **2** (219 mg, 0.621 mmol) and DIPEA (301 µL, 1.77 mmol) were successively added to a solution of G1 dendrimer **5** (215 mg, 0.156 mmol) in CHCl_3_ (20 mL), and the resulting mixture was refluxed for 5 days. After the reaction mixture was concentrated under reduced pressure, the residue was dissolved in CHCl_3_ (50 mL). The organic phase was washed with water (50 mL × 3), dried over Na_2_SO_4_, and concentrated under reduced pressure. The crude product was purified by silica gel column chromatography (gradient elution using CH_2_Cl_2_/EtOAc (2:1) until no detectable **2** was observed, as determined by UV spotting, to CHCl_3_/MeOH (10:1) to obtain the desired product) to afford **6** as an off-white solid (352 mg, 97%). ^1^H NMR (400 MHz, CDCl_3_) δ 7.38–7.30 (m, 12H), 7.26–7.22 (m, 6H), 5.48 (s, 6H), 5.30–5.20 (m, 3H), 4.99–4.89 (m, 3H), 4.68 (d, *J* = 6.1 Hz, 6H), 4.20 (dd, *J* = 5.7, 2.5 Hz, 6H), 3.78–3.68 (m, 60H), 3.49–3.40 (m, 12H), 2.21 (t, *J* = 2.5 Hz, 3H), 1.48 (s, 27H). ^13^C NMR (100 MHz, CDCl_3_) δ 166.1, 165.8, 165.4, 165.2, 154.8, 146.9, 134.6, 129.1, 128.7, 128.0, 121.5, 80.8, 79.9, 70.7, 54.1, 43.0, 43.0, 36.6, 30.5, 28.4. HRMS (MALDI): calcd for C_111_H_144_N_54_NaO_6_, 2352.2515 (M + Na^+^); found, 2352.2524.

### 3.6. Synthesis of G2 Dendrimer ***7*** from Compound ***6***

#### 3.6.1. CuSO_4_/Ascorbic Acid

Benzyl azide (16.5 µL, 0.132 mmol), ascorbic acid (10.4 mg, 0.0591 mmol), and copper (II) sulfate (1.9 mg, 0.0119 mmol) were successively added to a solution of compound **6** (93 mg, 0.0399 mmol) in THF/H_2_O (1:1 *v*/*v*, 5 mL). The resulting mixture was subjected to microwave irradiation at 110 °C for 30 min in a sealed vial or stirred at 110 °C for 2 h in a pressure vessel. After the reaction mixture was concentrated, the residue was dissolved in CHCl_3_ (70 mL), and the solution was washed with 5% aqueous NaOH (30 mL × 2) and water (30 mL × 2). The organic phase was dried over Na_2_SO_4_ and then evaporated. The crude product was purified by reprecipitation with MeOH from a clear solution of CHCl_3_ to afford **7** as an off-white solid. Yield; MW: (99.1 mg, 0.0363 mmol, 91%), pressure vessel: (92.2 mg, 0.0338 mmol, 85%).

#### 3.6.2. CuI/DIPEA

Benzyl azide (17.7 µL, 0.142 mmol), DIPEA (22 µL, 0.129 mmol), and copper (I) iodide (2.6 mg, 0.0137 mmol) were successively added to a solution of compound **6** (100 mg, 0.0429 mmol) in THF (5 mL). The resulting mixture was subjected to microwave irradiation at 110 °C for 30 min in a sealed vial or stirred at 110 °C for 2 h in a pressure vessel. After the reaction mixture was concentrated, the residue was dissolved in CHCl_3_ (70 mL) and the solution was washed with 5% aqueous NaOH (30 mL × 2) and water (30 mL × 2). The organic phase was dried over Na_2_SO_4_ and then evaporated. The crude product was purified by reprecipitation with MeOH from a clear solution of CHCl_3_ to afford **7** as an off-white solid. Yield; MW: (107 mg, 0.0392 mmol, 91%), pressure vessel: (106 mg, 0.0388 mmol, 91%). ^1^H NMR (600 MHz, CDCl_3_) δ 7.45–7.29 (m, 24H), 7.26–7.19 (m, 12H), 5.48 (d, *J* = 3.8 Hz, 12H), 5.35–5.15 (m, 6H), 4.67 (dd, *J* = 11.0, 6.0 Hz, 12H), 3.86–3.63 (m, 60H), 3.49–3.30 (m, 12H), 1.48 (s, 27H). ^13^C NMR (150 MHz, CDCl_3_) δ 166.1, 165.4, 165.2, 154.8, 146.9, 134.6, 129.1, 128.7, 128.0, 121.5, 79.9, 54.1, 43.1, 43.0, 42.9, 42.9, 36.6, 28.4. HRMS (MALDI): calcd for C_132_H_165_N_63_NaO_6_, 2751.4435 (M + Na^+^); found, 2751.4466.

### 3.7. Synthesis of G2 Dendrimer ***8***

TFA (10 mL) was added to a solution of compound **7** (467 mg, 0.171 mmol) in CH_2_Cl_2_ (10 mL), and the resulting mixture was stirred for 3 h at room temperature. After the reaction mixture was concentrated under reduced pressure, the residue was basified with 20% aqueous NaOH (30 mL). The aqueous solution was extracted with CHCl_3_ (30 mL × 5). The organic phase was washed with water (30 mL × 1), dried over Na_2_SO_4_, and concentrated under reduced pressure to afford **8** as a white solid (385 mg, 93%), which was used for the next reaction without further purification. ^1^H NMR (400 MHz, CDCl_3_) δ 7.44–7.28 (m, 24H), 7.26–7.19 (m, 12H), 5.47 (d, *J* = 2.6 Hz, 12H), 5.26 (t, *J* = 6.1 Hz, 6H), 4.67 (t, *J* = 5.5 Hz, 12H), 3.99–3.49 (m, 63H), 2.91–2.73 (m, 12H). ^13^C NMR (100 MHz, CDCl_3_) δ 166.1, 165.4, 165.2, 165.1, 147.0, 134.7, 134.6, 129.1, 128.7, 128.7, 128.0, 121.6, 121.5, 54.1, 46.0, 44.2, 43.1, 43.0, 36.6. HRMS (MALDI): calcd for C_117_H_141_N_63_Na, 2451.2862 (M + Na^+^); found, 2451.2861.

### 3.8. Synthesis of G3 Dendrimer ***9***

Compound **2** (85 mg, 0.241 mmol) and DIPEA (122 µL, 0.718 mmol) were successively added to a solution of G2 dendrimer **8** (145 mg, 0.0597 mmol) in CHCl_3_ (20 mL), and the resulting mixture was refluxed for 7 days. After the reaction mixture was concentrated under reduced pressure, the residue was dissolved in CHCl_3_ (100 mL). The organic phase was washed with water (50 mL × 2), dried over Na_2_SO_4_, and concentrated under reduced pressure. The crude product was purified by silica gel column chromatography (gradient elution using CH_2_Cl_2_/EtOAc (2:1) until no detectable **2** was observed, as determined by UV spotting, to CHCl_3_/MeOH (15:1) to obtain the desired product) to afford **9** as a pale yellow solid (161 mg, 80%). ^1^H NMR (400 MHz, CDCl_3_) δ 7.38–7.31 (m, 24H), 7.26–7.20 (m, 12H), 5.48 (d, *J* = 4.3 Hz, 12H), 5.26–5.16 (m, 6H), 4.68 (t, *J* = 5.9 Hz, 12H), 4.19 (dd, *J* = 5.8, 2.6 Hz, 6H), 3.92–3.65 (m, 84H), 3.47–3.40 (m, 12H), 2.20 (t, *J* = 2.5 Hz, 3H), 1.48 (s, 27H). ^13^C NMR (100 MHz, CDCl_3_) δ 166.2, 165.9, 165.4, 165.2, 165.2, 154.8, 146.9, 134.7, 129.1, 128.8, 128.1, 121.5, 80.9, 79.9, 70.7, 54.1, 43.0, 36.6, 30.5, 28.4. HRMS (MALDI): calcd for C_162_H_201_N_81_NaO_6_, 3399.7805 (M + Na^+^); found, 3399.7780.

### 3.9. Synthesis of G3 Dendrimer ***10*** from Compound ***9***

#### 3.9.1. CuSO_4_/Ascorbic Acid

Benzyl azide (11 µL, 0.0880 mmol), ascorbic acid (7.6 mg, 0.0432 mmol), and copper (II) sulfate (1.5 mg, 0.0094 mmol) were successively added to a solution of compound **9** (90 mg, 0.0266 mmol) in THF/H_2_O (1:1 *v*/*v*, 5 mL). The resulting mixture was subjected to microwave irradiation at 110 °C for 2 h in a sealed vial or stirred at 110 °C for 8 h in a pressure vessel. After the reaction mixture was concentrated, the residue was dissolved in CHCl_3_ (70 mL) and the solution was washed with 5% aqueous NaOH (30 mL × 2) and water (30 mL × 2). The organic phase was dried over Na_2_SO_4_ and then evaporated. The crude product was purified with column chromatography on silica gel (CHCl_3_/MeOH, 20:1) to afford **10** as a pale yellow solid. Yield; MW: (11.1 mg, 2.91 µmol, 11%), pressure vessel: (16.0 mg, 4.23 µmol, 16%).

#### 3.9.2. CuI/DIPEA

Benzyl azide (11 µL, 0.0880 mmol), DIPEA (14 µL, 0.0823 mmol), and copper (I) iodide (2.0 mg, 0.0105 mmol) were successively added to a solution of compound **9** (90 mg, 0.0266 mmol) in THF (5 mL). The resulting mixture was subjected to microwave irradiation at 110 °C for 2 h in a sealed vial or stirred at 110 °C for 8 h in a pressure vessel. After the reaction mixture was concentrated, the residue was dissolved in CHCl_3_ (70 mL) and the solution was washed with 5% aqueous NaOH (30 mL × 2) and water (30 mL × 2). The organic phase was dried over Na_2_SO_4_ and then evaporated. The crude product was purified with column chromatography on silica gel (CHCl_3_/MeOH, 20:1) to afford **10** as a pale yellow solid. Yield; MW: (14.0 mg, 3.71 µmol, 14%), pressure vessel: (28.2 mg, 7.46 µmol, 28%). ^1^H NMR (400 MHz, CDCl_3_:CD_3_OD = 5:1 *v*/*v*) δ 7.79–7.15 (m, 54H), 5.49 (d, *J* = 3.8 Hz, 18H), 4.81–4.48 (m, 18H), 4.41–4.16 (m, 9H), 3.92–3.56 (m, 84H), 3.45–3.35 (m, 12H), 1.49 (s, 27H). ^13^C NMR (100 MHz, CDCl_3_:CD_3_OD = 5:1 *v*/*v*) δ 165.8, 165.2, 164.9, 154.9, 146.8, 134.3, 130.9, 128.9, 128.6, 127.9, 121.8, 80.2, 65.5, 54.0, 42.8, 36.0, 29.5, 28.1, 25.4. HRMS (MALDI): calcd for C_183_H_222_N_90_NaO_6_, 3798.9725 (M + Na^+^); found, 3798.9722.

### 3.10. Synthesis of G1 Dendrimer ***11***

TFA (20 mL) was added to a solution of compound **3** (843 mg, 0.657 mmol) in CH_2_Cl_2_ (20 mL), and the resulting mixture was stirred for 3 h at room temperature. After the reaction mixture was concentrated under reduced pressure, the residue was basified with 20% aqueous NaOH (50 mL). The aqueous solution was extracted with CHCl_3_ (70 mL × 6). The organic phase was washed with water (30 mL × 1), dried over Na_2_SO_4_, and concentrated under reduced pressure to afford **11** as a white solid (555 mg, 86%), which was used for the next reaction without further purification. ^1^H NMR (400 MHz, CDCl_3_) δ 4.97 (t, *J* = 5.7 Hz, 3H), 4.19 (dd, *J* = 5.5, 2.5 Hz, 6H), 3.88–3.66 (m, 39H), 2.92–2.82 (m, 12H), 2.19 (t, *J* = 2.5 Hz, 3H). ^13^C NMR (100 MHz, CDCl_3_) δ 165.8, 165.4, 165.2, 165.1, 81.0, 70.6, 46.0, 44.3, 43.1, 43.0, 30.4. HRMS (MALDI): calcd for C_45_H_64_N_27_, 982.5833 (M + H^+^); found, 982.5820.

### 3.11. Synthesis of G2 Dendrimer ***12***

Compound **2** (775 mg, 2.20 mmol) and DIPEA (1.0 mL 5.91 mmol) were successively added to a solution of G1 dendrimer **11** (542 mg, 0.552 mmol) in CHCl_3_ (20 mL), and the resulting mixture was refluxed for 5 days. After the reaction mixture was concentrated under reduced pressure, the residue was dissolved in CHCl_3_ (100 mL). The organic phase was washed with water (50 mL × 2), dried over Na_2_SO_4_, and concentrated under reduced pressure. The crude product was purified by silica gel column chromatography (gradient elution using CH_2_Cl_2_/EtOAc (2:1) until no detectable **2** was observed, as determined by UV spotting, to CHCl_3_/MeOH (10:1) to obtain the desired product) to afford **12** as an off-white solid (936 mg, 88%). ^1^H NMR (600 MHz, CDCl_3_) δ 4.95 (t, *J* = 6.1 Hz, 6H), 4.25–4.15 (m, 12H), 3.92–3.70 (m, 60H), 3.49–3.41 (m, 12H), 2.21 (t, *J* = 2.5 Hz, 6H), 1.48 (s, 27H). ^13^C NMR (150 MHz, CDCl_3_) δ 165.8, 165.4, 165.2, 154.8, 80.9, 80.9, 79.9, 70.7, 43.1, 43.0, 42.9, 30.5, 28.4. HRMS (MALDI): calcd for C_90_H_123_N_45_NaO_6_, 1953.0595 (M + Na^+^); found, 1953.0604.

### 3.12. Synthesis of G2 Dendrimer ***7*** from Compound ***12***

#### 3.12.1. CuSO_4_/Ascorbic Acid

Benzyl azide (43.0 µL, 0.344 mmol), ascorbic acid (27.7 mg, 0.157 mmol), and copper (II) sulfate (5.3 mg, 0.0332 mmol) were successively added to a solution of compound **12** (101 mg, 0.0523 mmol) in THF/H_2_O (1:1 *v*/*v*, 5 mL). The resulting mixture was subjected to microwave irradiation at 110 °C for 30 min in a sealed vial or stirred at 110 °C for 2 h in a pressure vessel. After the reaction mixture was concentrated, the residue was dissolved in CHCl_3_ (70 mL) and the solution was washed with 5% aqueous NaOH (30 mL × 2) and water (30 mL × 2). The organic phase was dried over Na_2_SO_4_ and then evaporated. The crude product was purified by reprecipitation with MeOH from a clear solution of CHCl_3_ to afford **7** as an off-white solid. Yield; MW: (124 mg, 0.0454 mmol, 87%), pressure vessel: (126 mg, 0.0462 mmol, 88%).

#### 3.12.2. CuI/DIPEA

Benzyl azide (43.0 µL, 0.344 mmol), DIPEA (53 µL, 0.312 mmol), and copper (I) iodide (6.8 mg, 0.0357 mmol) were successively added to a solution of compound **12** (101 mg, 0.0523 mmol) in THF (5 mL). The resulting mixture was subjected to microwave irradiation at 110 °C for 30 min in a sealed vial or stirred at 110 °C for 2 h in a pressure vessel. After the reaction mixture was concentrated, the residue was dissolved in CHCl_3_ (70 mL) and the solution was washed with 5% aqueous NaOH (30 mL × 2) and water (30 mL × 2). The organic phase was dried over Na_2_SO_4_ and then evaporated. The crude product was purified by reprecipitation with MeOH from a clear solution of CHCl_3_ to afford **7** as an off-white solid. Yield; MW: (123 mg, 0.0451 mmol, 86%), pressure vessel: (129 mg, 0.0472 mmol, 90%).

### 3.13. Synthesis of G2 Dendrimer ***13***

TFA (15 mL) was added to a solution of compound **12** (193 mg, 0.100 mmol) in CH_2_Cl_2_ (15 mL), and the resulting mixture was stirred for 3 h at room temperature. After the reaction mixture was concentrated under reduced pressure, the residue was basified with 20% aqueous NaOH (50 mL). The aqueous solution was extracted with CHCl_3_ (100 mL × 5). The organic phase was washed with water (30 mL × 1), dried over Na_2_SO_4_, and concentrated under reduced pressure to afford **13** as a white solid (127 mg, 78%), which was used for the next reaction without further purification. ^1^H NMR (600 MHz, CDCl_3_) δ 4.87 (t, *J* = 5.8 Hz, 6H), 4.21 (td, *J* = 5.5, 2.5 Hz, 12H), 3.85–3.72 (m, 63H), 2.90–2.85 (m, 12H), 2.22–2.19 (m, 6H). ^13^C NMR (150 MHz, CDCl_3_) δ 165.9, 165.4, 165.3, 165.1, 81.0, 80.9, 70.7, 70.7, 46.1, 44.3, 43.1, 43.1, 43.0, 30.5. HRMS (MALDI): calcd for C_75_H_99_N_45_Na, 1652.9022 (M + Na^+^); found, 1652.9012.

### 3.14. Synthesis of Compound ***15***

Piperazine anhydrous (5.45 g, 63.3 mmol) was added to a solution of compound **2** (2.23 g, 6.32 mmol) in CHCl_3_ (30 mL), and the resulting mixture was stirred for 1 h at room temperature. After the reaction mixture was concentrated under reduced pressure, the residue was dissolved in CHCl_3_ (70 mL). The organic phase was washed with water (35 mL × 2), dried over Na_2_SO_4_, and concentrated under reduced pressure. The crude product was purified by column chromatography on NH silica gel (CHCl_3_/EtOAc, 20:1) to afford **15** as a white solid (2.01 g, 79%). ^1^H NMR (400 MHz, CDCl_3_) δ 5.01 (t, *J* = 5.8 Hz, 1H), 4.17 (dd, *J* = 5.6, 2.5 Hz, 2H), 3.85–3.61 (m, 9H), 3.48–3.38 (m, 4H), 2.93–2.79 (m, 4H), 2.19 (t, *J* = 2.5 Hz, 1H), 1.48 (s, 9H). ^13^C NMR (100 MHz, CDCl_3_) δ 165.8, 165.2, 165.0, 154.8, 80.9, 79.8, 70.6, 46.0, 44.2, 42.9, 30.4, 28.4. HRMS (MALDI): calcd for C_19_H_31_N_8_O_2_, 403.2565 (M + H^+^); found, 403.2565.

### 3.15. Synthesis of Compound ***16***

Cyanuric chloride (1.10 g, 5.99 mmol) and DIPEA (510.0 µL, 3.00 mmol) were successively added to a solution of compound **15** (1.20 g, 2.99 mmol) in CHCl_3_ (15 mL), and the resulting mixture was stirred for 2 h at 0 °C. After the reaction mixture was concentrated under reduced pressure, the residue was dissolved in CHCl_3_ (70 mL). The organic phase was washed with water (25 mL × 2), dried over Na_2_SO_4_, and concentrated under reduced pressure. The crude product was purified by column chromatography on silica gel (CHCl_3_/EtOAc, 15:1) to afford **16** as a white solid (1.60 g, 97%). ^1^H NMR (400 MHz, CDCl_3_) δ 4.94 (t, *J* = 5.7 Hz, 1H), 4.19 (dd, *J* = 5.7, 2.5 Hz, 2H), 3.95–3.68 (m, 12H), 3.49–3.40 (m, 4H), 2.20 (t, *J* = 2.5 Hz, 1H), 1.49 (s, 9H). ^13^C NMR (100 MHz, CDCl_3_) δ 170.4, 165.8, 165.2, 165.1, 164.1, 154.8, 80.6, 80.0, 70.8, 44.0, 43.0, 42.6, 30.5, 28.4. HRMS (MALDI): calcd for C_22_H_29_Cl_2_N_11_NaO_2_, 572.1775 (M + Na^+^); found, 572.1784.

### 3.16. Synthesis of Compound ***17***

Propargylamine (140 µL, 2.19 mmol) and DIPEA (1.24 mL, 7.29 mmol) were added to a solution of compound **16** (801 mg, 1.46 mmol) in THF (20 mL), and the resulting mixture was stirred for 1.5 h at room temperature. After the reaction mixture was concentrated under reduced pressure, the residue was dissolved in CHCl_3_ (70 mL). The organic phase was washed with water (50 mL × 3), dried over Na_2_SO_4_, and concentrated under reduced pressure. The crude product was purified by column chromatography on silica gel (CHCl_3_/MeOH, 20:1) to afford **17** as a white solid (708 mg, 85%). ^1^H NMR (400 MHz, CDCl_3_) δ 6.33–6.20 (m, 1H), 4.99 (t, *J* = 5.3 Hz, 1H), 4.26–4.14 (m, 4H), 3.93–3.70 (m, 12H), 3.48–3.41 (m, 4H), 2.23 (t, *J* = 2.5 Hz, 1H), 2.20 (t, *J* = 2.5 Hz, 1H), 1.48 (s, 9H). ^13^C NMR (100 MHz, CDCl_3_) δ 169.1, 165.8, 165.2, 165.2, 164.4, 154.8, 80.7, 79.9, 79.5, 71.2, 70.7, 43.4, 42.9, 42.7, 30.6, 30.5, 28.4. HRMS (MALDI): calcd for C_25_H_33_ClN_12_NaO_2_, 591.2430 (M + Na^+^); found, 591.2436.

### 3.17. Synthesis of G3 Dendrimer ***18***

Compound **17** (542 mg, 0.952 mmol) and DIPEA (502 µL, 2.95 mmol) were successively added to a solution of G1 dendrimer **5** (326 mg, 0.236 mmol) in CHCl_3_ (20 mL), and the resulting mixture was refluxed for 8 days. After the reaction mixture was concentrated under reduced pressure, the residue was dissolved in CHCl_3_ (100 mL). The organic phase was washed with water (30 mL × 1), dried over Na_2_SO_4_, and concentrated under reduced pressure. The crude product was purified by silica gel column chromatography (gradient elution using (CH_2_Cl_2_/EtOAc 2:1) until no detectable **16** was observed, as determined by UV spotting, to (CHCl_3_/MeOH 15:1) to obtain the desired product) to afford **18** as a slightly yellow solid (638 mg, 91%). ^1^H NMR (600 MHz, CDCl_3_) δ 7.41–7.29 (m, 12H), 7.26–7.22 (m, 6H), 5.49 (s, 6H), 5.32–5.23 (m, 3H), 5.17–4.84 (m, 6H), 4.69 (d, *J* = 6.1 Hz, 6H), 4.23–4.16 (m, 12H), 3.88–3.70 (m, 84H), 3.47–3.41 (m, 12H), 2.23–2.18 (m, 6H), 1.48 (s, 27H). ^13^C NMR (150 MHz, CDCl_3_) δ 166.1, 165.8, 165.4, 165.2, 154.8, 147.0, 134.6, 129.1, 128.8, 128.0, 121.5, 80.9, 80.9, 79.9, 70.7, 54.1, 43.0, 43.0, 36.6, 30.5, 28.4. HRMS (MALDI): calcd for C_141_H_180_N_72_NaO_6_, 3000.5885 (M + Na^+^); found, 3000.5873.

### 3.18. Synthesis of G3 Dendrimer ***10*** from Compound ***18***

#### 3.18.1. CuSO_4_/Ascorbic Acid

Benzyl azide (25.0 µL, 0.200 mmol), ascorbic acid (16.6 mg, 0.0943 mmol), and copper (II) sulfate (2.9 mg, 0.0182 mmol) were successively added to a solution of compound **18** (90 mg, 0.0302 mmol) in THF/H_2_O (1:1 *v*/*v*, 5 mL). The resulting mixture was subjected to microwave irradiation at 110 °C for 3 h in a sealed vial or stirred at 110 °C for 12 h in a pressure vessel. After the reaction mixture was concentrated, the residue was dissolved in CHCl_3_ (70 mL) and the solution was washed with 5% aqueous NaOH (30 mL × 2). The organic phase was dried over Na_2_SO_4_ and then evaporated. The crude product was purified with column chromatography on silica gel (CHCl_3_/MeOH, 20:1) to afford **10** as a yellow solid. Yield; MW: (12.3 mg, 3.26 µmol, 11%), pressure vessel: (12.6 mg, 3.33 µmol, 11%).

#### 3.18.2. CuI/DIPEA

Benzyl azide (25.0 µL, 0.200 mmol), DIPEA (31.0 µL, 0.182 mmol), and copper (I) iodide (4.2 mg, 0.0221 mmol) were successively added to a solution of compound **18** (90 mg, 0.0302 mmol) in THF (5 mL). The resulting mixture was subjected to microwave irradiation at 110 °C for 3 h in a sealed vial or stirred at 110 °C for 12 h in a pressure vessel. After the reaction mixture was concentrated, the residue was dissolved in CHCl_3_ (70 mL) and the solution was washed with 5% aqueous NaOH (30 mL × 2). The organic phase was dried over Na_2_SO_4_ and then evaporated. The crude product was purified with column chromatography on silica gel (CHCl_3_/MeOH, 20:1) to afford **10** as a pale yellow solid. Yield; MW: (11.0 mg, 2.91 µmol, 10%), pressure vessel: (18.1 mg, 4.79 µmol, 16%).

## 4. Conclusions

In this study, we conducted a fundamental investigation of click chemistry for melamine-based dendrimers. The click chemistry of G1 and G2 dendrimers containing three or six alkynyl groups with benzyl azide afforded the desired triazole dendrimers in high yields without undesirable byproducts, but some challenges were encountered in the case of G3 dendrimers. The desired reaction proceeded under microwave irradiation as well as with simple heating. This click chemistry can be utilized for various melamine dendrimers that are prepared with other amine linkers. Changing the piperazine linker may resolve the issues encountered here. In addition, this transformation may be used to achieve diverse functionalized azides with components showing medicinal activities, conjugation of biocompatible groups, and diagnostic labels. The stepwise methodology, which allows the repetition of divergent synthesis and click chemistry, can be suitable for introducing different triazole components. Further studies may allow the development of diverse functionalized melamine dendrimers.

## Data Availability

Data is contained within the article or Appendix A.

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
