# Peer review of "Click Chemistry of Melamine Dendrimers: Comparison of “Click-and-Grow” and “Grow-Then-Click” Strategies Using a Divergent Route to Diversity"

_molecules, 2022, doi:10.3390/molecules28010131_

Round 1
Reviewer 1 Report
This article by Yoshikazu Kitano et al. It is a solid research work where melamine dendrimers are treated in their two synthetic facets: click-and-grow and grow-and-click, where they encounter certain difficulties, especially given the insolubility of the compounds, but once resolved it will intensify. the value of methodology. All compounds (most with high molecular weight) are adequately characterized and it deserve publication in molecules ones minor questions have been addressed, such as:
- pg 2, line 55: instead of dendrimer 4 is 3.
- pg4, line123: reference 38 is not referenced.
- Even when they are related to the work, there are 14 citations out of 37 (38%) from one of the authors, they could be adequately treated to reduce the number since one includes the other.
Author Response
Thank you very much for reviewing our manuscript. Our reply to the report is bellow.
- pg 2, line 55: instead of dendrimer 4 is 3.
It was corrected.
- pg4, line123: reference 38 is not referenced.
- Even when they are related to the work, there are 14 citations out of 37 (38%) from one of the authors, they could be adequately treated to reduce the number since one includes the other.
Some references were deleted and reorganized.
Reviewer 2 Report
This paper reported the study of alkyne-azide click chemistry on melamine-based first-, second-, and third-generation triazine dendrimers. The click chemistry of the first- and second-generation containing three or six alkynyl groups with benzyl azide afforded the desired triazole dendrimers in high yields by simple reprecipitation without column chromatography, while those of the third-generation dendrimers encountered difficulties, such as low yields and poor solubility. The topic fits the scope of journal and may benefit the development of melamine dendrimers that are fabricated with various amine linkers. In general, the manuscript is well-organized and the experiments can support the conclusions. Some key issues are required to be addressed before its publication on Molecules.
1. In the introduction section, a figure illustration of previous work and current work in this study is suggested to be provided to improve the manuscript readability.
2. The purity of the target compounds are required to be tested.
3. The possible reasons (in mechanism) for the unsuccessful third-generation dendrimers are required to be discussed.
Author Response
Thank you very much for reviewing our manuscript. Our reply to the report is bellow.
- In the introduction section, a figure illustration of previous work and current work in this study is suggested to be provided to improve the manuscript readability.
This is the first report of alkyne-azide click chemistry on melamine-based dendrimer.
- The purity of the target compounds are required to be tested.
The purity was not tested by HPLC, but I guess it is more than 95% considering from NMR and MS.
- The possible reasons (in mechanism) for the unsuccessful third-generation dendrimers are required to be discussed.
As described in manuscript, I think the low solubility of the compounds prevented the synthesis of the third-generation dendrimers.